# Achalasia Subtype Differences Based on Respiratory Symptoms and Radiographic Findings

**DOI:** 10.3390/diagnostics13132198

**Published:** 2023-06-28

**Authors:** Jelena Jankovic, Branislava Milenkovic, Ognjan Skrobic, Nenad Ivanovic, Natasa Djurdjevic, Ivana Buha, Aleksandar Jandric, Nikola Colic, Jelena Milin-Lazovic

**Affiliations:** 1Clinic for Pulmonology, University Clinical Center of Serbia, 11000 Belgrade, Serbia; milenbra@gmail.com (B.M.); natalidjurdjevic@yahoo.com (N.D.); ivanabuha33@gmail.com (I.B.); jandricalexander@gmail.com (A.J.); 2Medical Faculty, University of Belgrade, 11000 Belgrade, Serbia; skrobico@gmail.com (O.S.); milinjelena@gmail.com (J.M.-L.); 3Clinic for Digestive Surgery, First Surgical University Hospital, Clinical Centre of Serbia, 11000 Belgrade, Serbia; nekic85@gmail.com; 4Center for Radiology and MR, University Clinical Center of Serbia, 11000 Belgrade, Serbia; drcola12@gmail.com; 5Institute for Medical Statistics and Informatics, University of Belgrade, 11000 Belgrade, Serbia

**Keywords:** achalasia, lung complications, aspiration, spirometry, chest CT

## Abstract

Three subtypes of achalasia have been defined using esophageal manometry. Several studies have reported that symptoms are experienced differently among men and women, regardless of subtype. All subtypes could have some impact on the appearance of respiratory symptoms and lung complications due to compression of the trachea or aspiration of undigested food. The aim of this research was to analyze the differences in respiratory symptoms and radiographic presentation of lung pathology depending on the diameter and achalasia types. One or more respiratory symptoms were reported in 48% of 114 patients, and all of them had two or more gastrointestinal symptoms. The symptom score (SS) is statistically significant for the prediction of subtype 1 (area under the curve = 0.318; *p* < 0.001, cut-off score of 6.5 had 95.2% sensitivity) and subtype 2 (area under the curve = 0.626; *p* = 0.020, cut-off score of 7.5 had 93.1% sensitivity). The most common type was subtype 2 (50.8%), and although only 14 patients had subtype 3, they had the largest esophageal diameter (mean 5.8 cm). The difference in esophageal diameter was significant between subtype 1 and 3 (*p* = 0.011), subtype 2 and subtype 3 (*p* = 0.011). Nine patients (6%) had mega-esophagus (four patients in type 1, three in type 2 and two in type 3). More than half of all patients (51.7%) had at least one parenchymal lung change on CT scan. Recurrent micro-aspirations led to changes in the structure of the airways and lung parenchyma such as ground glass (GGO) and nodular changes (12%) and fibrosis (14.5%), and they had higher esophageal diameters (*p* < 0.001). Patients with chronic lung CT changes had significantly higher esophageal diameter than with acute changes (*p* < 0.001). Awareness of the association of achalasia and lung disorders is important in early diagnosis and treatment. More than half (57.5%) of patients with achalasia had some clinical and/or structural pulmonary abnormalities. All three subtypes had similar respiratory symptoms, meaning they cannot be used to predict the subtype of achalasia; on the contrary, SS can predict the first two subtypes. A higher diameter of the esophagus is associated with chronic structural lung changes. Although unexpected, the pathological radiological findings and diameter were significantly different in subtype 3 patients, but those parameters cannot lead us to a specified subtype.

## 1. Introduction

Achalasia is an idiopathic motility disorder caused by insufficient lower esophageal sphincter (LES) relaxation. The lack of relaxation contraction of the LES is caused by degeneration of the myenteric plexus neurons. This disorder is characterized by aperistalsis, dilatation of the esophagus (or mega-esophagus), minimal LES opening and weak esophageal emptying of accumulated undigested contents into the stomach [1,2]. The golden standard for diagnosis of achalasia is manometry. This method can detect motility dysfunction of the esophagus and failure of relaxation of the LES [1,3]. Crespin and colleagues pointed out, as noted in many other studies, that patients with achalasia subtype 2 are the most common and have the best treatment outcomes [4]. The explanation for this is probably that subtype 1 is an end-stage phenotype with diminished myenteric plexus and subtype 3 is a spastic type [5].

The most common gastrointestinal symptoms in those patients are regurgitation, dysphagia of solid or liquid food, weigh loss, vomiting and chest pain [2]. Numerous studies have examined the connection between the subtype of achalasia, the dysphagia score and the frequency of symptoms. In Alvand et al.’s study, dysphagia and weight loss were most prevalent in subtype 3 and in women, while all other gastrointestinal symptoms were more common in subtype 2 [2].

Retention of food in the lower esophagus can often present as a feeling of chest pressure or retrosternal chest pain [6,7]. Usually, the pain is not too strong, is not constant and is said to be like “fire in the chest” or heartburning [8]. The mechanism of the pain can be linked to irritation of the acid content, compression of the airway and surrounding tissue or spasm of the esophagus itself. According to Laurino-Neto’s study, nearly 40% of patients experienced chest pain [9]. Patel and colleges found that chest pain was most frequently observed among patients with subtype 3 achalasia, probably because of esophageal spasms [10].

During embryonic development, the lungs appear as a blind evaporation of the anterior gyrus. Therefore, there is a risk of aspiration of ingested food from the digestive tract [6]. Because of that, in addition to the proximity of the esophagus to the trachea, there is a possibility that undigested food may be poured into the trachea and cause some respiratory symptoms, predominantly cough [11]. Usually, cough occurs at night, when the contents of the esophagus overflow irritate the respiratory mucosa, or after a meal [6,7]. Sometimes, that irritating cough that lasts for a few months or even years is the first symptom of the disease. During the diagnostic of the cause of a cough, after a bronchoscopy we can see only hyperemic mucosa of the respiratory tract (chemical irritation of mucosa), so gastroenterological examination is advised, and in many cases achalasia is detected. From our pulmonology experience, non-pulmonary causes of cough are gastroesophageal reflux, use of ACE inhibitors, post-nasal drip, mediastinal tumors such as lymphoma and many other reasons. Because of maladaptive lifestyle habits in individuals or ignoring symptoms such as prolonged cough, some patients are undiagnostic for a long time, which can be a reason for bad outcomes.

Structural pulmonary complications occur in more than half of all patients with achalasia. This can be due to recurrent micro- and macro-aspiration or tracheal compression from a dilated esophagus [9]. Aspiration consequently leads to damage to the respiratory endothelial cells and frequent respiratory superinfections, but also serious complications such as diffuse aspiration bronchiolitis, aspiration pneumonia, atelectasis, lung abscess and empyema. The reason is most likely a consequence of chemical damage to the alveolar–capillary membrane itself by acidic undigested content [12]. Lesions in the bronchial tree and lung parenchyma are primarily of a chemical nature. The severity of the clinical finding is determined by the amount of content and distribution in the lungs. The high acid content usually leads to a fatal outcome with the development of sepsis [12]. The question is: are some structural lung abnormalities connected with a specific achalasia subtype or esophageal diameter?

We must not ignore the impact of achalasia on the lung structure and function, or on the symptoms that patients have. Achalasia can be often misinterpreted as being of pulmonary origin. There are case reports in the literature of a misdiagnosis presented as acute respiratory distress syndrome or asthma [13]. However, in reality in those cases, the cause was aspiration of contents that overflowed from the esophagus in patients with achalasia. An important observation is that there is a lack of research on the influence of the diameter and type of achalasia on the symptom score and their consequences on the lung.

The aim of this research was to study the clinical and structural pulmonary abnormalities in patients with different types of achalasia and in correlation with the diameter of the esophagus.

## 2. Patients and Methods

### 2.1. Study Group

In this retrospective study, 114 patients who were surgically treated for achalasia in a clinic of digestive surgery in Serbia between 2014 and 2019 were included. This study was approved by the Ethics Committee in Belgrade and all subjects received and signed informed consent before enrollment (according to the principles of the International Declaration of Helsinki).

The diagnosis of achalasia was made on the basis of clinical symptoms, upper gastrointestinal endoscopy, chest CT scan and manometry [1]. Endoscopy findings were described as dilated esophagus, undigested food residue, resistance to the passage of the endoscope through the incomplete relaxation of the LES [1]. Diagnosis of achalasia was performed via manometry, with findings of loss of normal peristalsis and/or simultaneous contraction of the esophagus, high pressure and incomplete relaxation of LES [2]. Based on manometric pattern characteristics, achalasia was categorized in 3 different subtypes: subtype 1—with aperistalsis and the absence of panesophageal pressurization to more than 30 mm Hg, subtype 2—with intermittent periods of panesophageal pressurization, and subtype 3—with premature or spastic distal esophageal contractions [1].

### 2.2. Radiological Assessment

Chest X-ray radiography and chest CT scan were performed before surgery as preoperative preparations. Chest CT scan, as also an upper gastrointestinal endoscopy, was needed to exclude malignancies and other causes of patients’ symptoms which can look like achalasia. CT scans show the presence of pseudo-achalasia because of carcinoma, esophageal morphology, esophageal dilatation or lung structural changes. All CT structural findings were classified in two groups as acute (consolidation, bronchiolitis, lymphadenopathy) or chronic changes (fibro-adhesive finding, GGO, emphysema, nodulation, tracheal compression).

In charge of the evaluation of the CT findings were two radiologists with more than 10 years of experience in the field working in the Center for Radiology and NMR, University Clinical Center of Serbia.

### 2.3. Symptom Assessment

For the evaluation of the degree of gastroesophageal symptoms, this study used an extended version of the symptomatology questionnaire for achalasia—Vaesi’s symptom score as a modified Eskardt symptom score with a maximum total score of 15 [14]. It has 3 basic symptoms question (dysphagia, regurgitation and pain when swallowing). Every question was graded from 0 to 5, where: 0 is no symptoms; (1) is once a month or less often; (2) is once a week or 3–4 times monthly; (3) 2–4 is times a week; (4) is once a day; (5) is several times daily. Patients were interviewed on the initial visit. The interview was performed by a doctor and professor with more than 15 years of experience in the field working in the University Clinical Center of Serbia.

### 2.4. Statistical Analysis

Descriptive and analytical statistical methods were used in this study. the following descriptive methods were used: central tendency (arithmetic mean, median and mode), measures of variability (range, interquartile range and standard deviation) and relative numbers (structure indicators) for all study variables of interest. The results were presented as mean values (MV) with standard deviation (SD). Graphical and mathematical procedures were used to test the normality of distribution. Numerical data were compared with a Student t test for independent samples. Nominal data were compared with the Chi square test. A receiver operating characteristic (ROC) curve was used to determine cut-off values for continuous variables. The choice of the analytical statistical methods depended on the type of data and the distribution. Analyses were performed in IBM SPSS Statistics for Windows, version 22.0 (IBM Corp., Armonk, NY, USA, 2019). For figure presentation, we used MedCalc for Windows, version 19.4 (The statistical significance was set at *p* < 0.05.

## 3. Results

One hundred and fourteen operated patients (60 males and 54 female) with achalasia were included in this research. The mean age of the patients was 51.7 years (range 19–82 years). No statistically significant difference was found in the average age between the groups of patients with different types of achalasia (*p* = 0.565). The majority of patients had never smoked (53%); smokers made up nearly 30% patients, and others were ex-smokers.

Fifty-five percent of all patients had some comorbidity such as arterial hypertension, asthma, atrial fibrillation or diabetes. Those specific characteristics of patients were included in Table 1.

Using the manometry, achalasia subtype 1 was diagnosed in 42 (36.8%) of patients, subtype 2 in 58 (50.8%) and subtype 3 in only 14 (12.4%).

The mean diameter of the dilated esophagus was 5.4 ± 2.9 cm (range from 2.8 to 10.5), and the widest diameter was 10.5 cm (according to the chest CT scan). The mean width of the lumen of the esophagus in patients with subtype 1 was 4.8 cm, with subtype 2 it was 5.0, and in patients with subtype 3 it was 5.8. A statistically significant difference was found in the width of the esophagus between achalasia subtype 1 and subtype 3 (*p* = 0.009), as well as subtype 2 and subtype 3 (*p* = 0.011), while no statistically significant difference was found between subtypes 1 and 2. Nine of them had mega-esophagus (four patients in subtype 1, three in subtype 2 and two in subtype 3).

All patients had two or more gastrointestinal symptoms. Regurgitation was present in 41 (36%) and dysphagia in 92 patients (80.7%). There were statistically significant differences between symptoms. Dysphagia was the main symptom in subtype 1 and chest pain was main symptom in subtype 3. There was a significant association between the presence of regurgitation, dysphagia and dilatation of the esophagus, but without clinically significant differences in Vaesi’s symptom score between subtypes (Table 2).

The area under the curve of the symptom score in the prediction of subtype 1 and 2 achalasia was examined; the results are shown in Figure 1. The symptom score is statistically significant (area under the curve = 0.318; *p* < 0.001) for the prediction of subtype 1 achalasia. A cut-off score of 6.5 had 95.2% sensitivity for the diagnosis of subtype 1 achalasia. Additionally, the symptom score is statistically significant (area under the curve = 0.626; *p* = 0.020) for the prediction of subtype 2 achalasia. A cut-off score of 7.5 had 93.1% sensitivity for the diagnosis of subtype 2 achalasia.

One or more respiratory symptoms were seen in near half of all patients, and the most common was dry cough, mostly atby night, but the frequency of all respiratory symptoms is summarized in Table 3. Cough was the main respiratory symptom in subtype 1.

All patients were operated on laparoscopically in a clinic of digestive surgery in Serbia, and a myotomy was performed. For six of those patients, this was a reoperation for achalasia of an operation that was previously performed more than 12 years ago. Four of them underwent a classical abdominal myotomy, but two of them had pneumatic dilation. There were not any complications during or after surgery.

Nineteen patients had one kind or a combination of parenchymal lung changes on X-ray chest radiography. The most common lung CT scan manifestations were: 40 with ground glass opacification (GGO) and fibrosis, 9 with nodular changes, 8 with air trapping, 7 with consolidation, 7 with compression dilated esophagus on trachea and a small number with nodular changes, lymphadenopathy or tree-in-bud (Figure 2).

Esophagus diameter was compared according to CT; results are presented in Table 4.

Patients with pathological CT findings had significantly higher diameters. Additionally, patients with fibrosis and ground glass had significantly higher esophageal diameters. There was no significant difference according to acute CT changes. Patients with chronic changes had significantly higher esophageal diameters.

Types of achalasia were compared according to CT manifestations; results are presented in Table 5. Half of the patients in all three subtypes had pathological CT changes, without significant differences. Around 20% of all patients in the three types had CT fibrosis as the dominant finding, without significant differences between groups. Ground glass was predominant in subtype 3, but without significant differences. Acute changes in CT were present in five patients with subtype 1, six patients with subtype 2 and one patient with subtype 3, without significant differences. Chronic changes were dominant in subtype 3, but without significant changes between groups.

## 4. Discussion

Achalasia, as a gastrointestinal disorder, was first described by the doctor Sir Thomas Willis in 1674 [1]. Per the literature, achalasia is more common in patients between 30 and 60 years [3]. Less than 5% of cases were found before adolescence [13,15]. In our case, as well as the literature, most patients were older than 30 years; only nine patients in our study were adolescents, and there were none younger than 18 years.

Using high-resolution manometry, based on the Chicago classification version 4 from 2021, achalasia was categorized into three subtypes [16]. In our study, subtype 2 achalasia was reported to be predominant (more than half of patients) and the most rarely represented was subtype 3. Subtype 2 achalasia has also been reported as the most common subtype of achalasia in many other studies as well [17,18]. According to the study of Boeckxstaens and Zaninotto, subtype 3 occurs in only 10% of patients; similarly, in our study, it was rarest subtype, present in only 12% of all patients [19].

Knowing that the symptoms of achalasia are not only typical for this disorder, as we have already mentioned, justifies the diagnosis delay in detecting this disease, even up to 5 years after the appearance of the first symptom [1]. This leads to a loss of quality of life and to complications, such as mega-esophagus and even an esophageal carcinoma [1]. In our group, the average number of years from the onset of symptoms until diagnosis is 2.9 years, which is in accordance with the data from the literature. The longest duration of symptoms, 6.25 years on average, was seen in subtype 3, which is also correlated with the widest diameter, resulting in the appearance of chronic lung CT changes. The shortest duration of symptoms, on average 1.25 years, was seen in subtype 1, which is correlated with the appearance of dysphagia as the most dominant symptom of this subtype. Because of the burden and intensity of symptoms, these subtype 1 patients had more recently reported to the doctors. The patients gave their anamnestic data on the three main symptoms (dysphagia, regurgitation, retrosternal pain) when filling out the Vaezi symptom score (SS) questionnaire. All patients had two or more gastrointestinal symptoms. Regurgitation was present in one third of all patients and dysphagia in 92 patients in all three subtypes, though predominant in the first two subtypes. Retrosternal pain was predominant in subtype 3. The explanation for pain probably lies in the hypertrophic musculature along the distal part of the esophagus, compared to the other two subtypes achalasia, where the contractions are less strong and the esophagus wall is thinner [20]. Observing these three symptoms, no statistically significant differences can be seen in the SS between the three subtypes of achalasia. However, regurgitation and dysphagia were more dominant in two first subtypes, maybe because those two subtypes are characterized by aperistalsis and collection of food in the dilated esophagus (70% of our patients with mega-esophagus was in these two subtypes), while in subtype 3, contraction of the muscle layer pushes food through the cardia [20,21]. According to the definition given by Orringer and Stirling for mega-esophagus [22], nine patients were diagnosed with a diameter of more than 8 cm, with predominance in subtype 1.

Haft’s study is the first study to indicate that the SS is the gold standard for measuring the intensity of symptoms of achalasia [21]. The study by Tsuboi et al., which included 143 patients with achalasia, has proven that there is a high positive correlation and predictive value of SS with all subtypes of achalasia [23]. In our study, it was proven only in the first two subtypes, which may be caused by the small number of subjects with subtype 3 achalasia. SS shows a high sensitivity for predicting subtypes of achalasia, even at lower score values (6.5 in achalasia subtype 1 and 7.5 in achalasia subtype 2).

Almost half of all our patients had one or more respiratory symptoms. That is similar to the literature; according to Andolfi et al., the rate is about 41% [7]. The most common symptoms were in the form of cough or retrosternal pain.

A dilated esophagus is considered when a lumen width greater than 2.5 cm is observed. The average diameter of the dilated esophagus of our patients was 5.4 cm, and the widest diameter was 10.5 cm. The mean width of the esophageal lumen in patients with subtype 1 achalasia was 4.8 cm, with subtype 2 achalasia it was 5.0 cm and in patients with subtype 3 achalasia it was 5.8 cm. These data do not match the data from many studies, where a predominantly wider esophageal lumen was noted in subtype 1 [24,25]. The explanation for the literature-described dominance in the width of the lumen in subtype 1 is that in this subtype, aperistalsis is dominant due to the advanced stage of the disease and the accumulation of food and constant pressure on the esophageal wall, leading to dilatation. In this way, the wall of the esophagus is dilated and thinned [24,25]. While the low frequency of dilated esophagus in subtype 3, according to the literature, is explained by the fact that constant spasms of the esophagus do not allow rapid dilatation of the lumen, but, contrary to subtype 1, in this subtype of achalasia the wall is hypertrophic [22]. The data of our research are contrary to the previous data; a possible explanation for this is the small number of patients with subtype 3 achalasia (three times less than with subtype 1 and four times less than with subtype 2), so the interpretation of the overall results is difficult. Mega-esophagus was diagnosed in nine patients in our study group. According to the distribution in subgroups: four were patients with subtype 1, three were patients with subtype 2 and two were patients with subtype 3 achalasia. Our data on the predominance of frequency of mega-esophagus in subtype 1 achalasia coincide with the data of the research of Salvador and colleagues, as well as with the research of Meillier and others [18,24,25]. Although there is a limitation in the interpretation of the results due to the small number of subjects with subtype 3 achalasia, the obtained results are in correlation with the literature data of previously research.

In addition to the frequency of symptoms, we also evaluated the existence and frequency of structural lung changes in relation to diameter and achalasia subtype. Makharia and colleagues showed that 53% of patients with achalasia had structural lung abnormalities [26], similar to our study, where they were seen in 51.7% of our patients. Recurrent micro- and macro-aspirations lead to changes in the structure of the airways and lung parenchyma. If acute aspiration has occurred, consolidation or atelectasis can be seen [27]. Half of the study group had pathological findings. The most common lung CT scan manifestations were: 40 with GGO and fibrosis, 9 with nodular changes, 7 with consolidation, 7 with compression dilated esophagus on trachea and a small number with nodular changes, lymphadenopathy, atelectasis or tree-in-bud. Forty patients in this study already had GGO or fibrosis upon CT scan, which may indicate that the disease lasted longer, or that aspirations were chronic and extensive. That is consequence of chemical damage of the alveolar–capillary membrane with undigested content, which leads to permanent damage, so these changes remain chronic. Patients in all three subtypes had CT fibrosis as the dominant finding. Patients with chronic changes had significantly higher esophageal diameters. Chronic changes were dominant in patients with subtype 3, which also had higher esophageal diameter.

It is also important to judge whether, according to those CT scan findings, it is some acute pathological finding to treat before surgical treatment or if is chronic. All patients with consolidation, as the predominant acute preoperative CT manifestation, needed antibiotic preoperative treatment. In our population group, there was no significant difference in esophageal diameter according to acute CT changes. The limitation is the impossibility of interpreting the results and comparing between the subtypes, because there was only one patient in subtype 3. Seven patients had compression with a dilated esophagus on the trachea and those were patients with mega-esophagus. This can be a serious and potentially life-threatening complication [28]. Changes that could correspond to consolidation or atelectasis were seen radiographically in only 3 patients, while an acute inflammatory infiltrate such as pneumonia or tree-in-bud bronchiolitis was seen in 13 patients with a chest CT scan. They required immediate treatment with antibiotic therapy. Fibrosis and nodular changes can rarely be diagnosed in classic radiography, but they were described in 47 patients with achalasia using computed tomography.

As we can see, more than half (57.5%) of patients with achalasia have some functional and/or structural abnormalities in their lungs.

All 114 patients with achalasia included in this study were operated on laparoscopically at the Clinic for Digestive Surgery. They underwent Heller’s myotomy surgery. Surgical literature papers about LHM in combination with partial fundoplication indicate that this technique remains the gold standard. With this procedure, there is the possibility of adding an antireflux procedure—fundoplication—with which the morbidity rate is reduced, the duration of hospitalization is shorter and recovery is faster [29]. Laparoscopic Heller’s myotomy is a very successful method for treatment for patients with achalasia and the outcome does not vary significantly depending on the manometric subtype, according to a study by Crespin and colleagues, who examined the outcome of surgical treatment depending on the subtype of achalasia [4]. Constantini and a group of surgeons proved that LHM has an excellent final outcome, success and efficiency and that in this way symptoms can be alleviated in more than 80% of patients up to 20 years after surgery [30]. In a five-year follow-up of the outcome of operatively treated patients with Heller’s myotomy or after pneumatic dilation in 201 patients with achalasia by Moonen et al., it was proven that PD and LHM have a similar success rate. However, 25% of patients with PD require re-dilatation during follow-up or some other subtype of retreatment within the next 2 or more years, depending on the procedure and the stage of the disease [31]. For six of our patients, this was reoperative treatment of achalasia recurrences that had previously been treated surgically or nonsurgically an average of more than 12 years ago. Of the six reoperated patients (three patients with subtype 1 and three with subtype 2), two had previous pneumatic dilatation (subtype 1). In a study of 248 patients, Raja and colleagues investigated the frequency of symptoms and reinterventions after myotomy based on the subtype of achalasia. Patients undergoing myotomy should be aware that those with poorer esophageal function—achalasia subtype 1—are likely to require one or more postoperative reinterventions [32]. These data are consistent with our data on reoperated patients, in that the majority belonged to the group with subtype 1 achalasia whether PD or LHM was performed. All patients were discharged fully recovered after 3 to 5 days from the operation. This short period of hospitalization coincides with literature data on a shorter period of postoperative hospital stay [29].

There are several limitations in this study. It is a single center study with an average number of patients, as in other studies, because it is very rare disorder. Additionally, the small number of subtype 3 patients (as globally in other studies) and the most valid comparison with other groups is not possible. Using MDCT helps assess whether there is compression of the dilated esophagus on the trachea or heart, which can be misinterpreted as obstructive pulmonary disease, arrhythmia or GERD. It can alienate us doctors from the right diagnosis.

## 5. Conclusions

Pulmonary symptoms, as well as radiological and functional abnormalities, are common in patients with achalasia. More than half (57.5%) of patients with achalasia had some clinical and/or structural pulmonary abnormalities. All three subtypes had similar respiratory symptoms, meaning they cannot be used to predict the subtype of achalasia; on the contrary, SS can predict the first two subtypes. A higher diameter of the esophagus is associated with chronic structural lung changes. Although unexpected, the pathological radiological findings and diameter were significantly different in subtype 3 patients, but those parameters cannot lead us to a specified subtype. Awareness of the association of achalasia and lung disorders is important in early diagnosis and treatment. Therefore, advances in the knowledge of pulmonary manifestations in those patients through interaction within multidisciplinary teams would have benefits in diagnostics, treatment and outcome, and could impact prognosis and quality of life in this group of patients.

## Figures and Tables

**Figure 1 diagnostics-13-02198-f001:**
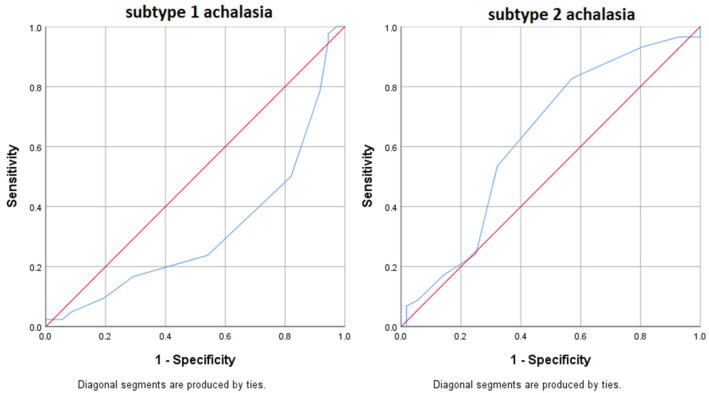
Area under the curve symptom score-achalasia subtype 1 and 2.

**Figure 2 diagnostics-13-02198-f002:**
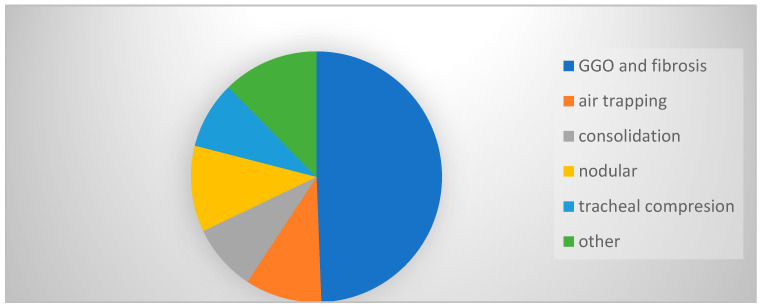
Chest CT findings.

**Table 1 diagnostics-13-02198-t001:** Specific characteristics of patients.

Gender	*n* (%)
Female/male	54 (47.4)/60 (52.6)
SMOKING STATUS	
•Smokers	34 (29.8)
Ex smokers	19 (16.7)
Non smokers	61 (53.5)
MOST COMMON COMORBIDITY	
•arterial hypertension	24 (21.0)
cardiac disease	12 (10.5)
asthma or COPD	11 (9.6)
atrial fibrilation/arrhythmya	7 (6.1)
hypothyroidism	7 (6.1)

**Table 2 diagnostics-13-02198-t002:** Vaesi’s symptom score and diameter according to subtypes of achalasia.

Type	1	2	3	*p*
	*n* = 42	*n* = 58	*n* = 14	
Symptom score	8.8 ± 1.9	9.7 ± 2.1	9.8 ± 2.5	0.075
Diameter (cm)	4.8 ± 1.7	5 ± 1.4	5.8 ± 2.3	0.169

**Table 3 diagnostics-13-02198-t003:** Clinical and functional pulmonary findings.

Respiratory Symptoms	*n* (%)
•cough	54 (47.3)
retro sternal pain	12 (10.5)
dyspnea	11 (9.6)
dysphonia	9 (7.9)
hoarseness of the voice	3 (2.6)

**Table 4 diagnostics-13-02198-t004:** Esophagus diameter according to CT changes.

			Mean ± SD	*p*
CT	59	pathological	5.6 ± 1.9	<0.001
	55	normal	4.3 ± 1	
CT fibrosis	92	no	4.7 ± 1.5	<0.001
	22	yes	6.2 ± 1.7	
CT GGO	96	no	4.7 ± 1.4	<0.001
	18	yes	6.5 ± 1.8	
acute	102	no	5 ± 1.6	0.524
	12	yes	5.3 ± 1.8	
chronic	67	no	4.2 ± 0.9	<0.001
	47	yes	6.1 ± 1.8	

**Table 5 diagnostics-13-02198-t005:** Subtypes of achalasia were compared according to CT changes.

Type		1	2	3	*p*
		*n* = 42	*n* = 58	*n* = 14	
CT	pathological	23 (54.8)	28 (48.3)	8 (57.1)	0.742
	normal	19 (45.2)	30 (51.7)	6 (42.9)	
CT fibrosis	no	33 (78.6)	47 (81)	12 (85.7)	0.838
	yes	9 (21.4)	11 (19)	2 (14.3)	
CT GGO	no	38 (90.5)	48 (82.8)	10 (71.4)	0.217
	yes	4 (9.5)	10 (17.2)	4 (28.6)	
acute	no	37 (88.1)	52 (89.7)	13 (92.9)	0.879
	yes	5 (11.9)	6 (10.3)	1 (7.1)	
chronic	no	24 (57.1)	36 (62.1)	7 (50)	0.687
	yes	18 (42.9)	22 (37.9)	7 (50)	

Data are presented as *n* (%).

## Data Availability

The data that support the findings of this study are available from the corresponding author (J.J.) upon reasonable request.

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
