# Peer review of "Achalasia Subtype Differences Based on Respiratory Symptoms and Radiographic Findings"

_diagnostics, 2023, doi:10.3390/diagnostics13132198_

Round 1
Reviewer 1 Report
Jankovic E and coworkers reported a case series of 114 patients with achalasia with mixed retrospective and prospective (not well defined) evaluation comparing the manometric (type 1,2 and 3), the Ekardt symptoms' score, the degree of esophageal dilatation and the chest complications at CT scan. They identified some statistical correlations among symptoms score and type of achalasia, as well esophageal diameters. In addition, more than half of patients had respiratory symptoms and/or lung complication with significant correlation with esophageal diameter.
Major
The study address a specific complication of (in general) long-lasting untreated achalasia such as respiratory complications and is interesting in this regards the evaluation of their frequency and characteristics.
What I find less interesting is the attempt to correlate the symptoms score to the Achalasia subtype. I don't see the rationale and interest there. More than the total score, some specific symptoms, i.e. chest pain may eventually be more correlated to type 3 for example. That an higher symptoms score correlates with the 2 more frequent subtypes is not surprising. The attempt should be to see whether a specific symptoms correlate more to type 1 or 2 but really I don't think so.
Similarly, it is somewhat expected the larger esophageal diameter of the esophagus may lead to higher risk of regurgitation and pulmonary issue. What has been completely ignored is the duration of symptoms in this context
Minor
The manuscript will definitively benefit from an English language revision and shortening.
Introduction - More than the contraction is the lack of relaxation of the LES the consequence of myenteric plexus degeneration. Manometry by definition measures pressure, no need to add "pressure manometry".
Need improvement
Author Response
Response to Reviewer 1 Comments
- We corrected definition of the type of study it is retrospective (line 104 in corrected version)
- More than half of our patients had respiratory symptoms. Frequency was presented in Table 3. Cough was the main respiratory symptom in subtype 1 (line 195). Other other symptoms were significantly less frequent.
- For suggestion „The attempt should be to see whether a specific symptoms correlate more to type 1 or 2“- There were statistically significant difference between symptoms. Dysphagia was the main symptom in subtype 1 and chest pain was main symptom in subtype 3 (line 177-179)
- For suggestion „duration of symptoms in this context.. “- The longest duratiron of symptoms 6,25 years on average, was in subtype 3, what is correlated with the widest diameter, resulting in the appearance of chronic lung CT changes. The shortest duration of symptoms ,on average 1,25 years was in subtype 1, wich is correlated with the appearance of dysphagia as the most dominant symptom of this subtype. Because of the burden and intensity of symptoms these subtype 1 patients, had more recently reported to the doctors (line 252-258).
- According to Your and the editor's suggestion, the discussion was shortened
- For "Introduction - More than the contraction is the lack of relaxation of the LES the consequence of myenteric plexus degeneration. Manometry by definition measures pressure, no need to add "pressure manometry"- we corrected definition and excluded word pressure.
Reviewer 2 Report
"Eckardt" symptom score. misspelled.
There are no parameters of the Chicago classification mentioned in this manuscript. Is there any association between the diameter of the dilated esophagus and the IRP level?
Author Response
Respected reviewer,
Thank you very much on your suggestions. We changed misspeled mistake Eckardt.
Unfortunately, we do not have high resolution manometry ,so we can not do IRP.
Reviewer 3 Report
Dear Editor,
I have reviewed the manuscript "Achalasia Subtype Differences based on Respiratory 2
Symptoms and Radiographic Findings". Jankovic et al analyzed the differences in respiratory symptoms and radiographic presentation depending on the diameter and achalasia types of lung pathology. Of 114 patients, 48% had one or more respiratory symptoms and all had 2 or more gastrointestinal symptoms. More than half (51.7%) had at least one parenchymal lung change on CT scan. Esophageal diameter was found to be significantly higher in patients with chronic lung CT changes compared to acute changes. Higher diameter of the esophagus is associated with chronic structural lung changes. Unexpectedly, the pathological radiological findings and diameter are significantly different in subtype 3 patients.
These are the findings known in the literature and seen in clinical experience. Unfortunately, I think the present study will not be of interest to the reader.
Best regards.
Moderate editing of English language
Author Response
Dear reviewer,
- Thank you for your opinion
- If you have any other suggestions to improve our work and make it more interesting for readers, we shall gladly accept them. Our idea was to show, based on the experience of our center how achalasia affects the respiratory tract and changes in the lungs. We think, perhaps, different experiences can contribute to a better understanding of this pathology.
Round 2
Reviewer 1 Report
Thanks for the revision